# The Construction and Validation of Nomogram to Predict the Prognosis with Small-Cell Lung Cancer Followed Surgery

**DOI:** 10.3390/cancers14153723

**Published:** 2022-07-30

**Authors:** Lei-Lei Wu, Wu-Tao Chen, Chong-Wu Li, Si-Hui Song, Shu-Quan Xu, Sheng-Peng Wan, Zhi-Yuan Liu, Wei-Kang Lin, Kun Li, Zhi-Xin Li, Dong Xie

**Affiliations:** 1Department of Thoracic Surgery, Shanghai Pulmonary Hospital, School of Medicine, Tongji University, Shanghai 200433, China; wull_7@yeah.net (L.-L.W.); lcwlcw55@163.com (C.-W.L.); 2010710@tongji.edu.cn (W.-K.L.); leeq8110@163.com (K.L.); 2School of Medicine, Shanghai Jiao Tong University, No. 227 South Chongqing Road, Shanghai 200025, China; ryzxcwt@126.com; 3School of Medicine, Tongji University, Shanghai 200092, China; 2031072@tongji.edu.cn (S.-H.S.); xsq2051070@163.com (S.-Q.X.); 2131046@tongji.edu.cn (S.-P.W.); 2131068@tongji.edu.cn (Z.-Y.L.)

**Keywords:** small-cell lung cancer, nomogram, survival, surgery, postoperative radiotherapy

## Abstract

**Simple Summary:**

The therapeutic effect of postoperative radiotherapy for small-cell lung cancer (SCLC) patients with limited stage remained unclear. The aim of this retrospective study was to construct and validate a nomogram to assess the prognosis of small-cell lung cancer patients followed surgery in a large cohort (882 patients) which involved patients from Shanghai Pulmonary Hospital and the Surveillance, Epidemiology and End Results database. The nomogram derived from the training cohort achieved good predictive ability and could discriminate low- and high-risk subgroups in four cohorts. Postoperative radiotherapy promoted survival for high-risk patients but had little effect on low-risk patients. Moreover, by subgroup analysis based on the N stage, we suggested that N2 patients in the high-risk subgroup could benefit more from postoperative radiotherapy. Therefore, our nomogram might help with clinical decisions on the strategy of postoperative radiotherapy for SCLC patients.

**Abstract:**

This study constructed and validated a prognostic model to evaluate the survival of small-cell lung cancer (SCLC) patients following surgery, and shed light on the strategy of postoperative radiotherapy. A total of 882 patients from Shanghai Pulmonary Hospital and the Surveillance, Epidemiology and End Results database after lung resection were selected. Multivariable Cox analysis was used to identify the indicators affecting long-term survival in patients. A nomogram was constructed to predict the prognosis of eligible patients. Indices of concordance (C-index) was used to access the predictive ability of cancer-specific survival (CSS) for the prognostic model. CSS discrimination in the prognostic model was comparable in the training and validation cohorts (C-index = 0.637[NORAD-T], 0.660[NORAD-V], 0.656[RAD] and 0.627[our hospital], respectively. Stratification based on the cutoff value of the nomogram yielded low- and high-risk subgroups in four cohorts. For patients in the high-risk group, postoperative radiotherapy was considered a survival-promoting strategy (unadjusted HR 0.641, 95% CI 0.469–0.876, *p* = 0.0046). In the low-risk group, however, the implementation of radiotherapy barely had an influence on CSS. In conclusion, the nomogram we constructed and validated could predict the prognosis of SCLC patients followed surgery and identify high-risk patients who were likely to benefit from postoperative radiotherapy.

## 1. Introduction

Lung cancer has still a leading death rate of malignant disease, although its incidence rate has decreased from the first to the second rank according to a recent report [1]. Small-cell lung cancer (SCLC), as a part of the pathological types in lung cancer, accounts for about 15% of cases in all lung cancers, and remains poor survival [2,3]. The 5-year overall survival of SCLC is only 6% [3]. Previous studies confirmed that SCLC patients who received operations had much better survival than patients without surgical treatment, especially in patients without lymph-node metastasis [4,5,6]. Thus, the guidelines of the national comprehensive cancer network recommend that SCLC patients with clinical stage I-IIA should receive lobectomy preferred [7]. Besides, adjuvant chemotherapy which is effective to control the progression of SCLC, can improve the survival outcomes and is recommended for conventional treatment by some guidelines from different regions [8,9].

For patients with clinical stage I-IIA, if their resected lymph nodes are confirmed as negative after the operation, the guidelines will not recommend prophylactic cranial irradiation or adjuvant localized chest radiotherapy [7,8]. However, a few studies found that adjuvant radiotherapy was associated with improved survival outcomes for some patients with clinical stage I-IIA [10,11]. Besides, for patients with pathological stage IIB-IIIA, the guidelines recommend adjuvant chemotherapy plus radiotherapy for patients with metastasis of ipsilateral mediastinal lymph node. However, the evidence of guidelines’ recommendation mainly is based on clinical trials which only enrolled patients without surgery, and there is still no consistent view on the decision of adjuvant radiotherapy for patients with metastasis of lymph nodes in the peri-bronchial or ipsilateral hilar [7,8]. Thus, it is unclear to make the treatment plan of adjuvant radiotherapy for the part of SCLC patients with limited stage. In addition, the evaluation of the prognoses for those patients remains important. Previous studies confirmed that nomograms had better performance in predicting prognosis than classic tumor, nodes, and metastasis (TNM) staging systems [12,13]. Therefore, this study aimed to construct a nomogram to improve the predictive ability for survival outcomes, and give clinical reference of adjuvant radiotherapy in postoperative SCLC patients with limited stage.

## 2. Materials and Methods

### 2.1. Patients

The Ethics Committee of Shanghai Pulmonary Hospital approved this study (K22-249), and the human data was in accordance with the Declaration of Helsinki in the manuscript. All of the cases were derived from the Surveillance, Epidemiology and End Results (SEER) database and Shanghai Pulmonary Hospital. In this study, TNM stages were reassigned according to the 8th American Joint Committee on Cancer (AJCC). Eligible patients met the following criteria: (1) pathologically diagnosed as SCLC; (2) complete follow-up; (3) patients with virtual survival status and clear survival time; (4) diagnosed between 2004–2015 for SEER database, and between 2014–2018 in Shanghai Pulmonary Hospital; (5) underwent surgery and the information about post-operative radiotherapy was clear. Patients were excluded if they: (1) were diagnosed with N3 or M1 diseases; (2) had unknown resected or positive lymph nodes; (3) had unknown tumor stage or surgery type (Figure 1). A total of 882 patients (SEER, N = 739, our hospital, N = 143) were collected, and patients from the SEER database were split into two groups, RAD (radiotherapy group, N = 245) and NORAD (no radiotherapy group, N = 494), based on their radiotherapy status. A training group (NORAD-T, N = 242) and a validation group (NORAD-V, N = 252) were further divided from the NORAD group randomly in a 1:1 ratio. Cases in our hospital and RAD were also used for validation.

### 2.2. Follow-Up

The median follow-up time was 51, 66, 45 and 60 months in NORAD-T, NORAD-V, RAD and our hospital respectively. The time interval between resection of the cancer and the cancer-caused death was defined as cancer-specific survival (CSS). Telephone interviews or office visits were adopted as follow-up methods. Those patients were actively followed up and conducted as a part of routine care.

### 2.3. Statistical Analysis

Categorical variables were compared using Pearson’s Chi-square test. Univariable and multivariable Cox regression analyses were performed to evaluate the prognostic significance of sex, age, race, marital status, tumor grade, tumor location, tumor size, surgery type, TNM stage, chemotherapy, lymph node ratio (LNR, the number of positive lymph nodes divided by the number of lymph nodes dissected), pathological T stage, and pathological N stage. A two-sided *p* < 0.05 was defined as statistically significant. Hazard ratios (HR) with 95% confidence intervals (95% Cis) were calculated by univariable and multivariable Cox proportional hazard regression analyses. Cases were censored at the end of follow-up. CSS was considered best with respect to clinical relevance. Nomogram was applied as an approach to demonstrate the prognostic significance of eligible variables [12,14]. After excluding confounding factors by multivariable analysis, we constructed the model based on four variables: sex, age, TNM stage, and LNR. The prognostic score (PS) was calculated through the nomogram for each case and was regarded as an indicator for CSS. Each group was further divided into high- and low-risk subgroups according to an optimal PS cutoff value. Survival curves were generated through Kaplan-Meier analysis and compared by log-rank tests. For evaluation of the prognostic model, indices of concordance (C-index) were generated for each dataset. In this study, the random classification of the NORAD group was conducted by the “scorecard” package in R 4.1.2 software [15]. The optimal cutoff point of PS and LNR was calculated by the “survminer” package in R software and X-tile 3.6.1 software (Yale University School of Medicine, New Haven, CT, USA) [16,17] respectively. The analyses were performed using R (version 4.1.2).

## 3. Results

### 3.1. Patient Characteristics

Table 1 presents the baseline characteristics of the SEER cohorts. In this study, women outnumbered men, constituting 55.1% of the patients. A total of 293 (39.6%) patients were age 65 and below, whereas 446 (60.4%) were over 65. The majority of patients were diagnosed with poorly or undifferentiated SCLC, comprising more than 93% of the patients. In terms of the TNM stage, most patients were diagnosed with stage IA SCLC. The proportion of patients who underwent chemotherapy was high, reaching 63.9%. Compared with NORAD-V, NORAD-T exhibited no statistical difference in baseline characteristics. However, there were differences between NORAD-V and RAD in the analysis of TNM stage, N stage, chemotherapy, LNR, and age. The baseline characteristics of our hospital was shown in Appendix A
Table A1.

### 3.2. Univariable and Multivariable Analysis

The outcome of the univariable and multivariable analysis was presented in Table 2. In order to discriminate the prognostic factors, a total of 11 variables were included in the univariable Cox regression analysis. Male sex (vs. female; *p* = 0.044), older age (>65 vs. ≤65; *p* = 0.014), tumor location (*p* = 0.012), TNM stage (*p* < 0.001), and LNR (*p* < 0.001) were considered to aggravate the survival. Other variables including race, marital status, grade, tumor size, surgical type, chemotherapy, and TNM stage had no significant influence on survival. Furthermore, multivariable analysis confirmed sex, age, TNM, and LNR as independent prognostic factors after eliminating confounding factors.

### 3.3. Development and Validation of the Nomogram

In order to predict the CSS of SCLC patients, we further visualized the result of multivariable Cox analysis using a nomogram (Figure 2). The calibration curves revealed that the predicted survival probability of the nomogram was close to the actual survival rate, indicating the good predicting accuracy of the nomogram (Figure A1). The C-index of NORAD-T was 0.637 (95% CI 0.630–0.645), which was acceptable (Figure 3A). In the internal validation cohort (NORAD-V), the C-index reached 0.660 (95% CI 0.654–0.667). For the RAD cohort and the cohort of our hospital, the C-index was 0.656 (95% CI 0.649–0.662), and 0.627 (95% CI 0.550–0.668), respectively, indicating a good discriminating ability (Figure 3 and Figure A2).

### 3.4. Risk Stratification Based on the Nomogram

In order to further explore the risk stratification potency of the nomogram, PS was generated by adding the score of each variable, which was listed in Figure 2B. An optimal cut-off value of PS was 122, which was determined through the “survminer” R package and thus divided the cohort into high- and low-risk groups. 64 patients in the NORAD-T cohort, and 52 patients in the NORAD-V cohort were categorized into the high-risk group. In all of four cohorts, high-risk groups defined by the nomogram showed significantly poorer survival than low-risk groups (NORAD-T: unadjusted HR 2.566, 95% CI 1.768–3.723, *p* < 0.001, NORAD-V: unadjusted HR 3.377, 95% CI 2.277–5.008, *p* < 0.001, RAD: unadjusted HR 2.062, 95% CI 1.431–2.970, *p* < 0.001, our hospital: unadjusted HR 2.183, 95% CI 1.225–3.888, *p* = 0.0061) (Figure 3).

### 3.5. Postoperative Therapeutic Options for Nnomogram-Defined Subgroups

Then, we integrated the high-risk and low-risk groups of the four cohorts respectively to explore the role of postoperative radiotherapy in patients of different risk statuses. There were 244 patients in the high-risk group and 638 patients in the low-risk group, respectively. For patients in the high-risk group, postoperative radiotherapy was considered a survival-promoting strategy (unadjusted HR 0.641, 95% CI 0.469–0.876, *p* = 0.0046) (Figure 3). The 3-year CSS of patients were 41.1% vs. 31.1% in the high-risk group (radiotherapy vs. no-radiotherapy). In the low-risk group, however, the implementation of radiotherapy had no influence on CSS (HR 0.956, 95% CI 0.723–1.266, *p* = 0.75). To further investigate the clinical significance of our nomogram, a sub-group analysis was performed. There were 169 patients with classification N2 diseases in the four cohorts. 92 of 169 patients underwent postoperative radiotherapy, although they did not receive survival benefits from it (Figure 4A). However, 68 patients with classification N2 diseases, in the high-risk group defined by our nomogram, had prognostic improvement attributable to postoperative radiotherapy (unadjusted HR 0.569, 95% CI 0.364–0.888, *p* = 0.01, Figure 4B). Patients with classification N2 diseases who belonged to the low-risk group and received postoperative radiotherapy did not have better survival than those patients without postoperative radiotherapy (Figure 4C). Postoperative radiotherapy did not improve the survival outcomes in patients with classification N1 as shown in Figure 4D. After classifying into the high-risk group by the nomogram, classification N1 patients with postoperative radiotherapy performed better prognostic trend over cases without this treatment, even though, the *p*-value was not significant statistically (Figure 4E). It was not meaningful for patients with classification N1 in the low-risk group, of note, to receive postoperative radiotherapy in order to reach satisfactory survival outcomes (Figure 4F).

## 4. Discussion

In this study, we used the SEER database to select the SCLC patients in accordance with our standards. Then, those patients were divided into three cohorts, a training group (NORAD-T), and two internal validation groups (NORAD-V and RAD). The data of patients in the NORAD-T was analyzed, and four indicators affecting prognosis were identified to develop a nomogram to predict the prognosis of SCLC followed surgery. Furthermore, this nomogram was validated in the NORAD-V, RAD, and the data of our hospital. Confirming the validity of the prognostic model in different cohorts indicates that the model is relatively stable and has applicability. To further explore the effectivity of the prognostic model in clinical practice, we used sub-group analysis based on the status of classification N. The results showed that classification N2 patients in the high-risk but not low-risk group could receive survival benefits from postoperative radiotherapy. These results may be helpful for clinicians to evaluate the prognosis of SCLC patients who followed surgery and give some reference information about treatment suggestions.

Postoperative radiotherapy is important for SCLC patients with lymph node involvement, especially for those in the high-risk group. In the guidelines of different regions, classification N2 patients are recommended to receive chemotherapy and radiotherapy after surgery in order to reach satisfactory survival outcomes [7,8]. The evidence of guidelines’ recommendation mainly is based on two meta-analyses [18,19]. The majority of clinical trials included in the above-mentioned two meta-analyses which compared the prognosis of patients with radiotherapy and cases without radiotherapy, only enrolled patients without surgery. Those meta-analyses revealed that the addition of radiotherapy combined with chemotherapy was better than chemotherapy alone in SCLC patients. However, the role played by postoperative radiotherapy in SCLC patients with limited stage is still unclear. Besides, two impactful major society consensuses were recently published to address this topic [20,21]. The consensus strongly recommended radiotherapy for SCLC patients with inoperable T1-2N0M0 or extensive stage. Regrettably, there is still no strong recommendation for adjuvant radiotherapy in postoperative SCLC patients due to lacking the high-level clinical practice evidence. A study by Urushiyama H et al. used a national database in Japan to analyze the information of 564 patients and found that the recurrence-free survival of SCLC patients with lymph node metastasis did not extend even when added the postoperative radiotherapy [22]. Another study by Liu WS et al., however, obtained adverse results, and found that postoperative radiotherapy could improve the survival of N1 + 2 patients [23]. Those above-mentioned studies from Japan and China, however, had a small sample size of patients with postoperative radiotherapy. The results of the present study were interesting. In all cohorts, postoperative radiotherapy did not have an impact on SCLC patients with N2 diseases, which was similar to the results of the study by Urushiyama H et al. After selection by our nomogram, the results, similar to Liu WS et al., showed that N2 patients in the high-risk group could receive survival benefit from postoperative radiotherapy. As for N2 patients stratified in the low-risk group, we found that they might not need to receive postoperative radiotherapy. Given that the SEER database lacked the location of postoperative radiotherapy, we used the data of our hospital to identify the radiotherapy location, and found the N2 patients of the high-risk group all received thoracic radiotherapy. Therefore, we suggested that postoperative radiotherapy, which might be thoracic radiotherapy, was a key treatment for N2 patients following surgery. Besides, it was not necessary for N1 patients to receive postoperative radiotherapy. Our results presented that N1 patients in the low-risk group might not have a need for postoperative radiotherapy as the study by Urushiyama H et al. confirmed. Cases of N1 diseases in the high-risk group after postoperative radiotherapy performed better survival trends than cases without radiotherapy, which indicated that N1 patients in the high-risk group were likely to receive prognostic benefits from postoperative radiotherapy, though not statistically confirmed. The reason for this phenomenon may be due to the insufficient sample size of patients in this subgroup. The part of those results in the present study was closed to the findings by Liu WS et al. Thus, we think that postoperative radiotherapy is appropriate for some patients with N1 diseases. However, further research is needed to confirm those results.

LNR could be an easy clinical tool to evaluate the prognosis of resected SCLC patients. Previous studies had confirmed that LNR could predict the prognosis of different malignant tumors, such as non-small-cell lung cancer, oral cavity cancer, and esophageal adenocarcinoma [12,24,25]. LNR in the present study showed the largest weight in our nomogram by analyzing the SEER database. In other words, LNR might be the most important factor that affected the prognosis of SCLC patients with limited stage in this study. In addition, the high level of LNR was associated with the poor prognosis of SCLC patients of the limited stage. As an independent indicator affecting prognosis, LNR could provide supplementary information about survival outcomes of SCLC patients besides the TNM stage.

There are some flaws in this study. First, although the data we used was from a large sample-size database, some important information was not detailed, such as the surgical procedure, and the location of radiotherapy, as we could not obtain it in the SEER database. LNR is mainly influenced by sample size and surgical procedure. Therefore, lacking the information about the above two factors may have an impact on results. Besides, the sample size of our hospital was relatively small, thus more cases will be collected in future research. Second, we did not further categorize the cases with postoperative radiotherapy because of the lacked information of radiotherapy location. Although the information on radiotherapy location in our hospital was detailed, the sample size was small. Third, Information relating to the clinical staging before surgery and nodal upstaging was absent, as the data of positron emission tomography-computed tomography, magnetic resonance imaging, and invasive mediastinal lymph node staging was not obtained. In addition, the number of preoperatively proven SCLC or accidentally found SCLC was not tracked in both databases. The lack of this part of data might obscure the real situation of the treatment modality for such patients. In the SEER database, the key information about the resection completeness affecting the decision on postoperative radiotherapy was not clear. Accordingly, we developed the prognostic model using the data without radiotherapy to reduce the effect of lacking information about resection completeness on final results. We hope that further research with the above-mentioned variables involved can yield more detailed results. Fourth, given this study belonged to a retrospective study, thus, it was impossible to avoid selection bias. In this study, the distribution of TNM stage, LNR, and chemotherapy were not balanced between groups, which might affect the results of survival analyses. SCLC patients with advanced stage tended to be considered for radiotherapy, which resulted in a high proportion of advanced-stage patients being classified into the RAD group after grouping. Thus, more studies are necessary to further validate our findings.

## 5. Conclusions

A nomogram to predict the prognosis of SCLC patients followed surgery is constructed and validated. Postoperative radiotherapy is likely to improve the survival outcomes in the high-risk group.

## Figures and Tables

**Figure 1 cancers-14-03723-f001:**
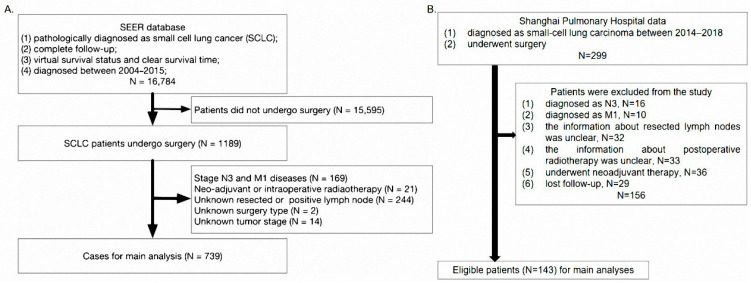
The flow chart of this study. SEER: Surveillance Epidemiology and End Results.

**Figure 2 cancers-14-03723-f002:**
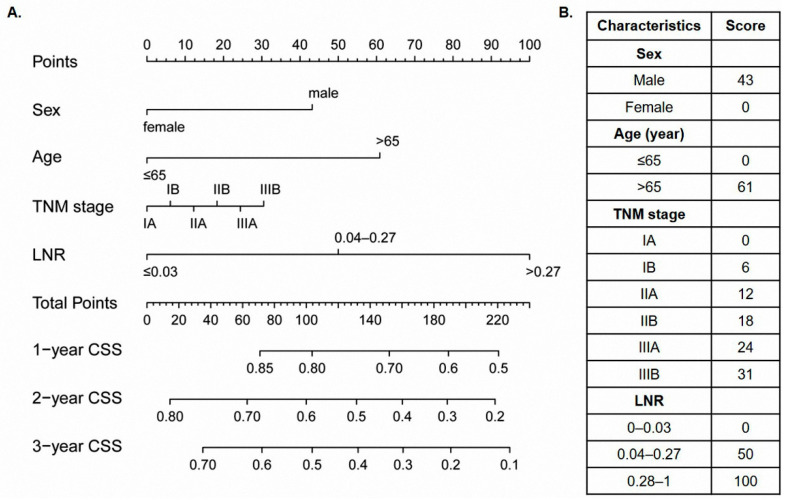
Postoperative prognostic nomogram for small-cell lung cancer patients (**A**) and the point assignment based on the nomogram (**B**).

**Figure 3 cancers-14-03723-f003:**
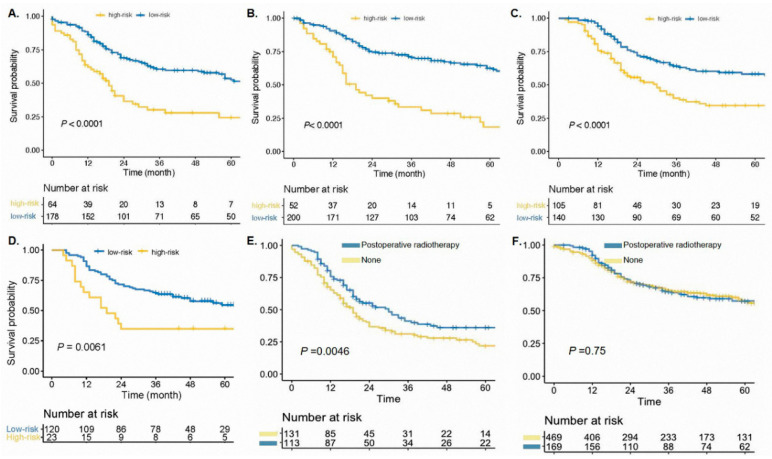
Cancer-specific survival curves for small-cell lung cancer patients after surgery according to the prognostic model in the NORAD-T (**A**), NORAD-V (**B**), RAD (**C**) and our hospital (**D**). Cancer-specific survival curves for those patients according to the postoperative radiotherapy in the high-risk group (**E**) and low-risk group (**F**). NORAD: no radiotherapy group, RAD: radiotherapy group.

**Figure 4 cancers-14-03723-f004:**
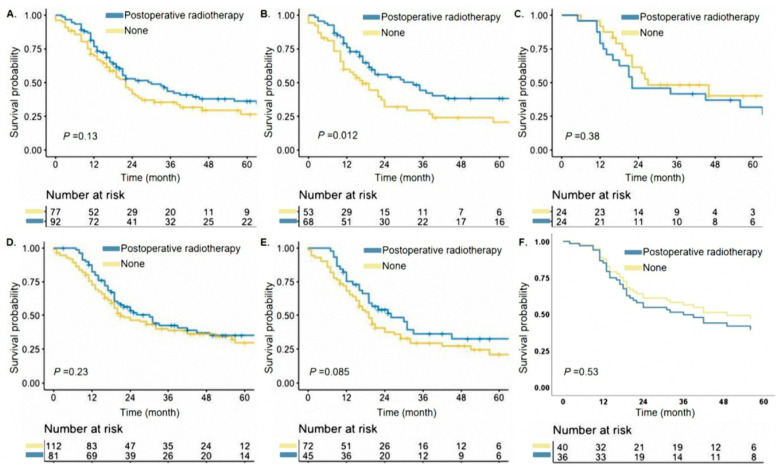
Cancer-specific survival curves for small-cell lung cancer patients after surgery according to the postoperative radiotherapy in the N2 cases (**A**), N2-high-risk cases (**B**), N2-low-risk cases (**C**), N1 cases (**D**), N1-high-risk cases (**E**) and N1-low-risk cases (**F**).

**Table 1 cancers-14-03723-t001:** Patient characteristics from the surveillance, epidemiology, and end results database.

Cohort	NORAD-T	NORAD-V	RAD	*p* *	*p* **
Total	242	252	245		
Sex (%)				0.094	0.718
Female	122 (50.4)	147 (58.3)	138 (56.3)		
Male	120 (49.6)	105 (41.7)	107 (43.7)		
Age (%)				0.280	0.018
≤65	79 (32.6)	95 (37.7)	119 (48.6)		
>65	163 (67.4)	157 (62.3)	126 (51.4)		
Race (%)				0.840	0.677
White	224 (92.6)	231 (91.7)	228 (93.1)		
Others	18 (7.4)	21 (8.3)	17 (6.9)		
Marital status (%)				0.233	0.122
Married	134 (55.4)	133 (52.8)	149 (60.8)		
Unmarried	95 (39.3)	112 (44.4)	87 (35.5)		
Unknown	13 (5.4)	7 (2.8)	9 (3.7)		
Grade (%)				0.372	0.239
I & II	18 (7.4)	20 (7.9)	12 (4.9)		
III	109 (45.0)	128 (50.8)	118 (48.2)		
IV	115 (47.5)	104 (41.3)	115 (46.9)		
TNM (%)				0.749	<0.001
IA	95 (39.3)	115 (45.6)	58 (23.7)		
IB	46 (19.0)	48 (19.0)	18 (7.3)		
IIA	13 (5.4)	11 (4.4)	7 (2.9)		
IIB	47 (19.4)	44 (17.5)	66 (26.9)		
IIIA	37 (15.3)	31 (12.3)	83 (33.9)		
IIIB	4 (1.7)	3 (1.2)	13 (5.3)		
N stage (%)				0.586	<0.001
N0	170 (70.2)	183 (72.6)	98 (40.0)		
N1	43 (17.8)	46 (18.3)	77 (31.4)		
N2	29 (12.0)	23 (9.1)	70 (28.6)		
Tumor size (%)				0.637	0.166
0–30 mm	166 (68.6)	183 (72.6)	176 (71.8)		
31–50 mm	62 (25.6)	59 (23.4)	48 (19.6)		
51–70 mm	10 (4.1)	6 (2.4)	13 (5.3)		
>70 mm	4 (1.7)	4 (1.6)	8 (3.3)		
Tumor location (%)				0.883	0.524
Upper lobe	145 (59.9)	157 (62.3)	137 (55.9)		
Lower lobe	72 (29.8)	74 (29.4)	82 (33.5)		
Middle lobe	17 (7.0)	14 (5.6)	18 (7.3)		
Others	8 (3.3)	7 (2.8)	8 (3.3)		
Surgery type (%)				0.449	0.063
Lobectomy	181 (74.8)	200 (79.4)	172 (70.2)		
Pneumonectomy	11 (4.5)	8 (3.2)	11 (4.5)		
Wedge	50 (20.7)	44 (17.5)	62 (25.3)		
LNR (%)				0.927	<0.001
0–0.03	173 (71.5)	184 (73.0)	99 (40.4)		
0.03–0.27	36 (14.9)	35 (13.9)	68 (27.8)		
0.27–1	33 (13.6)	33 (13.1)	78 (31.8)		
Chemotherapy (%)				0.706	<0.001
No	128 (52.9)	128 (50.8)	11 (4.5)		
Yes	114 (47.1)	124 (49.2)	234 (95.5)		

NORAD: no radiotherapy group, RAD: radiotherapy group, TNM: tumor, nodes and metastasis, LNR: lymph node ratio. * *p*-value represents the difference between NORAD-T and NORAD-V ** *p*-value represents the difference between NORAD-V and RAD.

**Table 2 cancers-14-03723-t002:** Univariate and multivariate analysis for mortality of small-cell lung cancer patients in the NORAD-T group.

Variable	Univariate Analysis		Multivariate Analysis	
	HR (95%CI)	*p*-Value	HR (95%CI)	*p*-Value
Sex		0.044		0.024
Female	Reference		Reference	
Male	1.450 (1.010–2.080)	0.044	1.540 (1.058–2.239)	0.024
Age		0.014		0.008
≤65	Reference		Reference	
>65	1.640 (1.090–2.480)	0.014	1.781 (1.164–2.726)	0.008
Race		0.590		
White	Reference			
Others	0.825 (0.403–1.690)	0.590		
Marital status		0.266		
Married	Reference			
Unmarried	0.878 (0.605–1.270)	0.493		
Unknown	0.483 (0.176–1.320)	0.157		
Grade		0.468		
I & II	Reference			
III	1.570 (0.716–3.460)	0.259		
IV	1.560 (0.711–3.420)	0.268		
TNM stage		<0.001		0.035
IA	Reference		Reference	
IB	1740 (1.050–2.890)	0.031	1.810 (1.090–3.005)	0.022
IIA	0.772 (0.274–2.170)	0.623	0.944 (0.332–2.682)	0.914
IIB	2.100 (1.290–3.430)	0.003	1.489 (0.681–3.254)	0.319
IIIA	2.100 (1.230–3.610)	0.007	1.388 (0.627–3.072)	0.418
IIIB	7.180 (2.540–20.300)	<0.001	5.539 (1.530–20.055)	0.009
Tumor size		0.430		
0–30 mm	Reference			
31–50 mm	1.390 (0.924–2.080)	0.115		
51–70 mm	1.280 (0.519–3.170)	0.590		
>70 mm	1.510 (0.476–4.790)	0.485		
LNR		<0.001		0.005
0–0.03	Reference		Reference	
0.03–0.27	1.460 (0.892–2.400)	0.132	1.128 (0.504–2.521)	0.770
0.27–1	3.190 (2.010–5.060)	<0.001	2.708 (1.249–5.869)	0.012
Tumor location		0.012		
Upper lobe	Reference			
Lower lobe	1.450 (0.974–2.160)	0.068		
Middle lobe	1.510 (0.794–2.870)	0.209		
Others	3.950 (1.790–8.710)	<0.001		
Surgery type		0.778		
Lobectomy	Reference			
Pneumonectomy	1.290 (0.598–2.790)	0.513		
Wedge	1.090 (0.705–1.690)	0.691		
Chemotherapy		0.132		
No	Reference			
Yes	0.758 (0.528–1.090)	0.132		

NORAD: no radiotherapy group, TNM: tumor, nodes and metastasis, LNR: lymph node ratio.

## Data Availability

Any researchers interested in this study could contact us for requiring the data.

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
