# Peer review of "The Construction and Validation of Nomogram to Predict the Prognosis with Small-Cell Lung Cancer Followed Surgery"

_cancers, 2022, doi:10.3390/cancers14153723_

Round 1

Reviewer 1 Report

I would like to congratulate the authors of an interesting article on the treatment of small cell lung cancer.

In their study, conducted on a very large group of patients, the authors analyzed the long-term results of SCLC surgery. The main aim of the study was to identify a group of patients in whom the use of postoperative radiotherapy may improve cancer-specific survival.

The results of the study are very interesting for lung cancer physicians and may be of clinical significance.

The manuscript is divided into appropriate sections and subsections, each containing all the relevant elements that should be included in it.

I only have a few comments.

1.       The manuscript is written in good quality English, but a revision by a English-editing service could improve its quality. This would be particularly important given the quality of the Journal and the potential relevance of the study conducted by the authors.

2.       In this type of study, information on the quality of preoperative staging would be useful, including the percentages of patients who underwent PET-CT, MRI of the brain and invasive mediastinal lymph node staging. Authors should also include information on the clinical stage before surgery and nodal upstaging. If this type of information is not available I suggest to include it in the Limitations subsection.

3.       The authors should include the information on the completeness of surgery. It would be interesting to analyze the outcomes of postoperative radiotherapy considering the completeness of resection (complete vs incomplete vs uncertain). If this type of information is not available I suggest to include it in the Limitations subsection.

Once again, I would like to congratulate the authors of a very interesting study. Although the changes suggested above could reveal some interesting aspects of qualification for postoperative radiotherapy, I believe that the study should also be published if it is not possible to introduce them. I suggest publication of the manuscript in the Cancers after minor revision.

Author Response

Dear professor,

Thanks for your timely review and professional suggestions.

Q1: The manuscript is written in good quality English, but a revision by an English-editing service could improve its quality. This would be particularly important given the quality of the Journal and the potential relevance of the study conducted by the authors.

Response: Thanks for your comments. We have invited a native speaker to check the article in aspects of grammar and fluency. We have screened our article and corrected minor grammar mistakes. Meanwhile, we have rephrased some sentences to make the language more fluent.

Q2: In this type of study, information on the quality of preoperative staging would be useful, including the percentages of patients who underwent PET-CT, MRI of the brain and invasive mediastinal lymph node staging. Authors should also include information on the clinical stage before surgery and nodal upstaging. If this type of information is not available, I suggest to include it in the Limitations subsection.

Response: Thanks for your professional comments. Unfortunately, the above-mentioned information was not accessible in the database. Clinical staging can influence the choice of treatment plans and nodal upstaging serves as a quality measure for the completeness of the nodal dissection, indicating the accuracy of N stage. The evaluation of these two variables could enhance the integrity of the study design and will be involved in further research. We have made changes in the discussion section (Page 9, line 334-401).

Q3: The authors should include the information on the completeness of surgery. It would be interesting to analyze the outcomes of postoperative radiotherapy considering the completeness of resection (complete vs incomplete vs uncertain). If this type of information is not available, I suggest to include it in the Limitations subsection.

Response: Thanks for your comments. It would be an interesting point to dig in to see the correlation of completeness of surgery with the effect of postoperative radiotherapy if only we had tracked the information, which was also a factor influencing the prognostic model. Thus, we developed the prognostic model using the data without radiotherapy to reduce the effect of lacking the information about resection completeness on final results. We hope that we will construct a well-organized study that including the variable in the future. We have mentioned this flaw in the discussion section (Page 9, line 404-407).

Reviewer 2 Report

To Authors: In the present paper Authors developed a prognostic score and a related nomogram for resected SCLC, that was validated in different populations of patients receiving or not adjuvant radiotherapy. The major finding is the efficacy of this prognostic score to identify high risk patients, those might benefit from adjuvant radiotherapy. The study is well designed, aims and methods are clearly exposed. Furthermore, on my knowledge this is one of the largest series focusing on resected SCLC.

I only have a major concern about the study grouping: the study population was divided in four groups, three from the SEER database and one from the Authors’ institutional database. Two groups from SEER were randomly created including patients who didn’t received adjuvant radiotherapy and were used as training and validation groups. The third group from SEER was used to test the nomogram in a population of patients who received adjuvant radiotherapy. I can’t find the reason of performing the same test on the fourth group, that is a heterogeneous group including both RT and NO-RT patients with a different chemotherapy rate as compared with the previous groups. I guess it would be better to move the RT-patients from this group to the RAD group, in order to have a single larger RT-group, reducing the number of groups/analyses (and tables) and making the paper more easily readable.

Minor revisions:

- Materials and Methods section: please specify how the diagnosis of SCLC was obtained: how many patients undergoing surgery for preoperatively proven SCLC and how many “surprise” SCLC did you included?

- Table1: Authors reported in the text some results from the N1 and N2 subgroups but table1 is lacking about the N status distribution. Please add the N status in table1.

- Table1: RAD group seems to include more advanced cancers. Please comment on this in the discussion.

- page 8 line 190: reporting six CSS rates in a row might be confusing, I would report the 3-year rates for the two groups only.

- Discussion: both the first and the second paragraph report comments on the relationship between the prognostic model (high risk vs low risk) and the benefit of RT in the N1 and N2 subgroups. I would eliminate all the lines between 221 and 235, moving this part of the discussion to the end of the second paragraph, where the analysis from the literature about the efficacy of radiotherapy in these subgroups is reported. Please also avoid to comment on the p-value of 0.085 of N1 subgroup, it would better to state that in N1-patients the efficacy of RT was not confirmed, regardless the risk classification.

Author Response

Dear professor,

Thanks for your constructive suggestions and careful review.

Q1: I only have a major concern about the study grouping: the study population was divided in four groups, three from the SEER database and one from the Authors’ institutional database. Two groups from SEER were randomly created including patients who didn’t received adjuvant radiotherapy and were used as training and validation groups. The third group from SEER was used to test the nomogram in a population of patients who received adjuvant radiotherapy. I can’t find the reason of performing the same test on the fourth group, that is a heterogeneous group including both RT and NO-RT patients with a different chemotherapy rate as compared with the previous groups. I guess it would be better to move the RT-patients from this group to the RAD group, in order to have a single larger RT-group, reducing the number of groups/analyses (and tables) and making the paper more easily readable.

Response: We appreciate your meticulous suggestion on the study design. As you mentioned, the fourth group, namely the cohort of “our hospital”, is a heterogeneous group with a different distribution of radiotherapy, and chemotherapy along with other basic characteristics. However, in order to testify the nomogram derived from the SEER database, patients from “our hospital” can best serve as an external validation group. In our study, the nomogram could discriminate between high- and low-risk patients in this cohort, showing good applicability when externally validated. Moreover, by introducing patients from “our hospital,” we could acquire a large increase in the sample size when performing the analysis of the integrated high- and low-risk groups as well as subgroup analysis based on the N stage, which was hard if only RT-patients were introduced. Thanks for your suggestion again, sincerely.

Q2: Materials and Methods section: please specify how the diagnosis of SCLC was obtained: how many patients undergoing surgery for preoperatively proven SCLC and how many “surprise” SCLC did you included?

Response: Thanks for your comments. The diagnosis of SCLC was pathologically determined in our hospital, as mentioned in Page 2, line 82. Neo-adjuvant radiotherapy was also excluded in the cohort (Figure 1). Nevertheless, it is a pity that the above-mentioned information was absent from the SEER database and our hospital, which may have an impact on the treatment selected. We have added this absence of information in the discussion sector (Page 10, line 401-402). We are sorry for it.

Q3: Table1: Authors reported in the text some results from the N1 and N2 subgroups but table1 is lacking about the N status distribution. Please add the N status in table1.

Response: Thanks for your comments. N status has been added in Table 1 as you suggested (Page 4, Table 1).

Q4: Table1: RAD group seems to include more advanced cancers. Please comment on this in the discussion.

Response: Thanks for your comments. As you mentioned, RAD group consists a higher proportion of IIB-IIIB tumors than NORAD. SCLC patients with advanced stage tended to be considered for radiotherapy, which resulted in a high proportion of advanced-stage patients being classified into the RAD group after grouping. We hope the problem of disproportion of variable can be tackled in the future study with a larger sample size. We have added changes accordingly in the discussion section (Page 10, line 410-414).

Q5: page 8 line 190: reporting six CSS rates in a row might be confusing, I would report the 3-year rates for the two groups only.

Response: Thanks for your comments. The sentence has been rephrased to present 3-year CSS rates only (Page 8, line 193).

Q6: Discussion: both the first and the second paragraph report comments on the relationship between the prognostic model (high risk vs low risk) and the benefit of RT in the N1 and N2 subgroups. I would eliminate all the lines between 221 and 235, moving this part of the discussion to the end of the second paragraph, where the analysis from the literature about the efficacy of radiotherapy in these subgroups is reported. Please also avoid to comment on the p-value of 0.085 of N1 subgroup, it would better to state that in N1-patients the efficacy of RT was not confirmed, regardless the risk classification.

Response: Thanks for your professional comments. As you pointed out, we found that our former arrangement of the article resulted in redundancy. Therefore, we have rearranged our sentences in the discussion section (Paragraph 1-2) according to your suggestion.